# LSTM-Autoencoder Based Anomaly Detection Using Vibration Data of Wind Turbines

**DOI:** 10.3390/s24092833

**Published:** 2024-04-29

**Authors:** Younjeong Lee, Chanho Park, Namji Kim, Jisu Ahn, Jongpil Jeong

**Affiliations:** 1Department of Smart Factory Convergence, Sungkyunkwan University, 2066 Seobu-ro, Jangan-gu, Suwon-si 16419, Republic of Korea; ioioiiopop@g.skku.edu (Y.L.); chanho124@g.skku.edu (C.P.); namji@skku.edu (N.K.); js3053@skku.edu (J.A.); 2AI Research Center, 20 Pangyo-ro, Bundang-gu, Gfyhealth, Seongnam-si 13488, Republic of Korea

**Keywords:** wavelet packet transform, high pass filter, principal component analysis, LSTM Autoencoder, unsupervised learning

## Abstract

The problem of energy depletion has brought wind energy under consideration to replace oil- or chemical-based energy. However, the breakdown of wind turbines is a major concern. Accordingly, unsupervised learning was performed using the vibration signal of a wind power generator to achieve an outlier detection performance of 97%. We analyzed the vibration data through wavelet packet conversion and identified a specific frequency band that showed a large difference between the normal and abnormal data. To emphasize these specific frequency bands, high-pass filters were applied to maximize the difference. Subsequently, the dimensions of the data were reduced through principal component analysis, giving unique characteristics to the data preprocessing process. Normal data collected from a wind farm located in northern Sweden was first preprocessed and trained using a long short-term memory (LSTM) autoencoder to perform outlier detection. The LSTM Autoencoder is a model specialized for time-series data that learns the patterns of normal data and detects other data as outliers. Therefore, we propose a method for outlier detection through data preprocessing and unsupervised learning, utilizing the vibration signals from wind generators. This will facilitate the quick and accurate detection of wind power generator failures and provide alternatives to the problem of energy depletion.

## 1. Introduction

At present, mankind is facing environmental and energy issues, of which the shortage in energy supply due to climate change has emerged as a serious problem. Meanwhile, countries worldwide have continued to increase their energy production and use to meet their economic and social demands. Moreover, while traditional non-renewable energy sources, such as oil and coal are being depleted, global countries are facing a decline in energy supply due to increasing energy demands [1]. Accordingly, renewable energy has attracted attention as an alternative energy source to solve these problems [2]. Renewable energy sources are advantageous because they release less carbon dioxide emissions than fossil fuels. In addition, renewable energy as a relatively new industry could create a positive effect on society by providing continuous job opportunities and aiding in economic development [3]. Moreover, with increasing interest in environmental and energy issues, the renewable energy industry is growing rapidly and is attracting global attention from numerous governments and companies that are expanding research, development, and investment in this sector [4].

Wind power generators are a representative means of renewable energy generation and are attracting attention as alternatives to global warming and energy resource depletion [5]. The advantages of wind power generators are as follows. Clean energy production is possible, and wind power generators do not generate air pollutants because they use wind to generate electricity without combustion reactions [6]. Thus, the depletion of energy resources is expected to be alleviated. Unlike energy generation by conventional fossil fuels, wind power generates electricity from natural energy sources. Thus, the problem of energy resource depletion can be alleviated. They can also be used to produce electricity sustainably. Wind power generators generate electricity using wind, which is constantly present on Earth; therefore, sustainable electricity production is possible. In the future, wind generators are expected to develop further and power generation will increase further. Existing wind power plants contain large machines with high power generation capacities and many restrictions on installation sites, but wind power generation is expected to become more efficient because of various technologies, such as miniaturization, advancement, and wind power as a medium [7].

Additionally, wind power generation will play an important role as an environmentally friendly means of power production, even in the face of increasing power demands. However, the availability, reliability, and turbine life of wind energy must be improved to compete with other energy sources. Organizational dynamics require strict cost management as the industry grows [8]. Currently, the problem of wind energy generation failure detection is largely divided into two. The first is accuracy and reliability, and the second is cost. First, it is difficult to measure the sensor due to the vibrations or noise generated by wind power generators, and accurate failure detection is difficult due to a lot of noise. In addition, wind generators must accurately classify each failure situation because various failure phenomena can occur depending on the environmental conditions. Second, for fault detection, many sensors must be installed, data must be collected, and complex algorithms are required. This is expensive for large-scale wind power plants and is difficult to apply in small-scale power plants. The larger the motor, the more frequent the failure; therefore, more attention is required. As the size and number of wind power generators increase, reducing monitoring and repair costs has become increasingly important [9].

Recent work on outlier detection includes [10] used a Kalman filter to generate appropriate residuals evaluated with a predetermined threshold and proposed an adaptive system. Deep learning networks such as data-driven convolutional neural networks (CNN) and stacked denoising autoencoders (SDAE) were also used but did not consider the hidden temporal dependencies of each parameter [11]. Reference [12] proposed an anomaly detection model combining long short-term memory (LSTM) and bidirectional long short term memory (BLSTM) to prevent failures caused by resource contention in cloud environments and demonstrated good experimental results. While existing anomaly detection methods require human intervention and computational complexity, making it difficult to apply in real-world scenarios, [13] showed a lightweight algorithm that can detect anomalies in real time and respond immediately. In [14], the maintenance mode of high-speed rail power equipment is updated with a predictive and proactive approach to accurately predict future maintenance times and solve the problem of data shortage. On the vision side, proactive anomaly detection (PAAD) based on planned movements and current observations is proposed to proactively detect anomalous behaviors in the navigation of mobile robots [15]. A mission-critical services (MCS) overload detection architecture is proposed to improve reliability and self-configuration by predicting service requirements and providing overload notifications in real-time through intelligent connectivity with 5G core networks [16]. In this study, we performed unsupervised learning using a long short-term memory autoencoder (LSTM-AE) with temporal characteristics considering only the normal vibration data of wind turbines. We compared the unsupervised learning methods of isolated forests and LSTM-AE to determine which model is better suited for unsupervised learning outlier detection.

The contributions of this study are as follows:The objective of this study was to diagnose faults using unsupervised learning based on time series vibration data in a noisy environment.In this study, we used wavelet packet decomposition to decompose one signal into multiple bandwidth signals and used a high-frequency filter to emphasize the high-frequency part. We used the filtered signal through principal component analysis to benefit from computing resources and analysis through dimensionality reduction.When applying the preprocessing method proposed in this study, the reconstruction error of learning and predicting the data by unsupervised learning was about 97%, which was better than the reconstruction error (about 94%) when no preprocessing method was previously applied.The preprocessing method proposed in this study has shown that it improves performance even for simple models. The results of this study are expected to be useful in signal and speech processing and will be helpful for new preprocessing methods.

The remainder of this paper is organized as follows: in Section 2, we introduce the technology used for the proposed data preprocessing. Section 3 includes the model structure of the anomaly detection proposed in this study. In Section 4, we describe the experimental results and evaluate the methods used. In Section 5, we conclude and outline implications for follow-up studies.

## 2. Related Work

### 2.1. Wind Power Generators

Wind power generators typically operate in complex environments. The collected vibration signals are typically mixed with non-fixed strong background noise. Therefore, it is difficult to extract useful features from the raw vibration signals. Additionally, the gearbox is combined with other mechanical parts of the wind turbine, which inevitably results in various inherent vibration modes, and the vibration signal exhibits multi-scale characteristics [17]. Wind power generators change their rotational speeds rapidly under changing weather conditions and complex industrial environments [18]. Therefore, the gearbox of a wind power generator is an important component [19]. The gearbox in the wind generator changes the direction and speed of the rotational motion and transmits it to the generator. However, the gearbox, which is the core of a wind power generator, is a component that can frequently fail due to fatigue and wear. Therefore, abnormality detection in gearboxes is essential for the safe and efficient operation of wind power generators [11]. Abnormality detection can prevent failures or accidents by monitoring various conditions inside the gearbox, such as vibration, temperature, current, and noise, and through quick response, can extend the life of the wind generator; thus, reducing maintenance costs.

### 2.2. Preprocessing Methods

The wavelet packet transform (WPT) shown in Figure 1 allows the signal to be decomposed and reconstructed into several signal components with the same bandwidth but different center frequencies. Using the characteristics of wavelet transmission has the advantage of providing higher accuracy in the high-frequency part of the signal and having no reconnection or missing information. WPT is suitable for handling non-fixed mechanical vibration signals with high-frequency characteristics and strong background noise in complex industrial environments. WPT is frequently used in feature extraction steps, which focus on manually extracting more useful features [20]. WPT can obtain multiscale signals by decomposing non-fixed signals, which is insufficient for CNN [17]. WPT has excellent positional properties in both the time and frequency domains and is suitable for compression. Furthermore, proper selection of the wavelet function allows the same signal to be represented with smaller errors [21]. WPT is a well-known mathematical tool that has special advantages over the Fourier transformation in the analysis of unfixed signals because it provides an arbitrary time-frequency resolution with redundant base features. Moreover, WPT is a more advanced version of the continuous wavelet transformation that can create a suitable level of resolution for each signal. Wavelet packets divide the frequency axis into finer pixel parts than the discrete wavelet transform (DWT). In wavelet packet decomposition, the discrete signal passes through a larger number of filters than in DWT [22].

In the wavelet analysis, signals are divided into detailed and approximate information. However, in wavelet packet analysis, we can distinguish the details and approximations that provide a good range for signal processing. Wavelet decomposition is used to solve the high-frequency components in a short time, whereas the low-frequency components require a significant amount of time. Here, low-frequency components consume large time intervals, whereas high-frequency components circulate at much shorter intervals [23]. Therefore, slowly changing components can only be identified at long time intervals, whereas rapidly changing components can be identified at short time intervals. Wavelet decomposition can be treated as a continuous-time wavelet decomposition sampled at different frequencies at each level or scale.

Principal component analysis (PCA) performs a dimensionality reduction on the input dataset while maintaining the characteristics of the dataset that contribute the most to the variance. PCA identifies patterns in data and expresses them in a new form by highlighting similarities and differences. PCA is used to find patterns in high-dimensional data and to overcome the limitations of visualization. Therefore, the PCA is a powerful tool for data analysis. The mathematical technique used in PCA is known as intrinsic analysis which calculates the eigenvalues and eigenvectors of the square-symmetric matrix using the sum of the squares and externalities. The eigenvectors associated with the largest eigenvalues correspond to the orientation of the first principal component, and those associated with the second-largest eigenvalues determine the orientation of the second principal component. The sum of the eigenvalues is the same as the trace of the matrix, and the maximum number of eigenvectors is the same as the number of rows (or columns) of the matrix. PCA is one of the simplest multivariate analysis methods that reveal the internal structure of data in a manner that best describes their variance [24]. When a multivariate data set is visualized as a coordinate set along each variable in a high-dimensional data space, the PCA can provide the user with a “shadow” of the data converted into a low-dimensional picture. This is the most useful way to express the core perspective of the data.

PCA is performed in a way that reduces the dimensions of the converted data using only a few main components. As PCA extracts the main components that best represent the characteristics of the data, the converted data are expressed in low dimensions while preserving the information of the original data as much as possible. Thus, PCA greatly aids in data analysis and visualization by reducing the dimensions of the data. In addition, PCA helps understand the structure of data by considering the correlations between variables [25]. Overall, PCA is a powerful tool for identifying patterns in data and highlighting similarities and differences to represent data in new forms. Therefore, it was used to reduce the dimensions of the data and to reveal its internal structure. PCA plays an important role in data analysis and visualization and provides meaningful information by extracting key data characteristics.

A high-pass filter (HPF) is one of the filters used in signal processing to block signals in the low-frequency region [26]. HPF is used to highlight or analyze high-frequency signals. Through this, the frequency area of the desired signal can be selected, and unnecessary low-frequency components removed. The operation source of HPF is small, and the signal of the low-frequency area is preserved. HPF is used in various applications, for example, in music and audio processing, or in image processing where it is used to extract high-frequency information such as boundary detection. In addition, signal processing may analyze the signal of the desired frequency band. There are several types of HPF, and various forms depending on the frequency response characteristics or design methods. HPF is one of the important filters used to analyze high-frequency signals. In the process of removing other components, the characteristics of the original signal are removed [27].

### 2.3. LSTM

LSTM is one kind of recurrent neural network (RNN) that has been widely used for time-series predictions and has been proven to perform well for long-term and short-term predictions. The LSTM network includes several gates for tracking the sequence of information and as shown in Figure 2 it comprises three gates: an input, a forgetting, and an output gate. The input gate selects the data to be stored for subsequent calculations. The forgetting gate selects data that are not stored in the state data. For example, a calculation result of zero indicates that information is fully preserved, whereas a calculation result of 1 indicates that the information was completely discarded. The output gate determines the state information to be routed to the output. *C* indicates the following conditions [28]. The LSTM cells use gate mechanisms to learn long-term dependencies. This mechanism adjusts the state of the memory through the input and deletes gates to ensure that important information can be maintained for a long time. This allows LSTM to learn and remember meaningful patterns, even in long sequences. LSTM has been used in various applications. Natural language processing uses it for sentence generation, machine translation, and emotion analysis. LSTM identifies long dependencies within sentences, helps to understand the context, and generates natural sentences. LSTM is also used to process time-series data such as speech recognition, music generation, and stock price prediction. In summary, the LSTM cells are a type of RNN, which is a proposed model for learning long-term dependencies. LSTM controls the flow of information through input, deletion, and output gates, and processes sequence data by maintaining a memory state. This allows the LSTM to perform well on data with long dependencies and is utilized in a variety of applications [29].

Figure 3 is the LSTM-AE model. LSTM-AE compresses and reconstructs the input data using LSTM cells. An autoencoder is an unsupervised learning model that learns the compression representations of the input data and reconstructs the input data based on them [30]. The LSTM-AE consists of an encoder and a decoder for the LSTM cell. The input data are compressed by the encoder and converted into a representation of the latent space. The decoder then receives a representation of the latent space as the input and reconstructs the original data [31]. The LSTM-AE learns to minimize the difference between the reconstructed and original data. Because LSTM cells can learn long-term dependencies, they can efficiently compress and reconstruct data by considering time flows [30]. This allows the LSTM-AE to extract important characteristics of the data and improve the reconstruction of noisy data. The LSTM-AE can be used in various applications, such as the dimension reduction of data, characteristic extraction, and anomaly detection. For example, using an LSTM-AE for anomaly detection in time-series data can detect strange behaviors or events using models that have learned normal patterns. Therefore, the LSTM-AE is an autoencoder based on LSTM cells that can be used in various applications by compressing and reconstructing time-series data. The LSTM-AE was utilized for anomaly detection using the reconstruction error by calculating the difference from the original data when the LSTM-AE reconstructed the input data and considered this large difference as an outlier [29]. The LSTM-AE performed model learning using only normal data during the learning stage.

The model learned normal patterns and adjusted the parameters to minimize reconstruction errors. Subsequently, the reconstruction error was calculated by inputting new data into the model. When the reconstruction error was high, it could be determined that the corresponding data showed an operation different from the normal pattern learned by the model [32]. These data could be considered as outliers. The reconstruction error was calculated by measuring the difference between the original and reconstructed data [33]. Generally, it is calculated using the square or absolute value of the error. When the error value was greater than a predefined threshold, the corresponding data were classified as outliers [34]. For outlier detection, the LSTM-AE utilized the time-series characteristics of the data. The LSTM cells could learn temporal dependencies and detect outliers through reconstruction errors. Accordingly, the LSTM-AE could detect outliers in the time-series data and nonlinear characteristics. Therefore, the LSTM-AE can be used to detect outliers using reconstruction errors and to detect and analyze outliers in time-series data by utilizing the ability of LSTM cells.

### 2.4. Anomaly Detection

Anomaly detection typically involves learning normal patterns from a given dataset using machine learning algorithms, and detecting strange behaviors or abnormal patterns in new data based on this learned pattern [35]. Anomaly detection is used in several areas, for example, in industrial areas to detect anomalies occurring in manufacturing processes and in security areas to detect malicious behavior such as intrusion or fraud. Anomaly detection typically involves the classification of new data by using models learned using only normal data. Models that learn from normal data typically identify the distribution of normal patterns and consider the data divided from this distribution as outliers [36]. Therefore, anomaly detection focuses on detecting outliers in data. Anomaly detection can be categorized into three types: supervised, semi-supervised, and unsupervised [37]. Supervised Anomaly Detection uses both labeled normal and abnormal data during learning, which requires an intuitive model of performance evaluation, solves class imbalance problems, and is more accurate than other methods [38]. Therefore, it is primarily used when high accuracy is required, and the more diverse the abnormal samples, the higher the performance. However, because of the nature of the manufacturing industry, abnormal data are not generated. Therefore, class imbalance problems are frequently encountered, and various studies, such as data augmentation, loss function redesign, and batch sampling, are being conducted to solve this problem [39]. Considering the high cost and lengthy time in obtaining these abnormal samples, if the class imbalance is severe Semi-Supervised Anomaly Detection is utilized to use only labeled normal data for learning. This can be attempted in situations in which it is difficult to collect abnormal data [40].

The key to this methodology is to establish a discriminant boundary surrounding the normal samples and narrow this boundary as much as possible to regard all samples outside the boundary as abnormal. One-class SVM is a representative methodology that uses one-class classification [41]. Unsupervised Anomaly Detection assumes that most of the data are normal when learning. This is easy when there is no definite normal labeling in the data, and a threshold setting for the distinction between the normal and abnormal data may be required after model learning. The simplest way to achieve this is to perform PCA on a given dataset to detect abnormal samples by reducing and restoring the dimensions. An autoencoder-based methodology is mainly used as a neural network-based methodology. The autoencoder proceeds with encoding, which compresses the input into code or late variables, and decoding, which restores it close to the original, allowing only important information of the data to be compressed [42].

### 2.5. Isolation Forest

Isolation forest is one of the algorithms for anomaly detection. This algorithm uses a tree-based segmentation method to detect outliers in the data [43]. The Isolation Forest operates as follows. First, an attribute is randomly selected from the dataset, and an arbitrary division value is selected between the minimum and maximum values of the attribute. The data are divided into two subgroups based on the selected segmentation value. The same segmentation process is repeated recursively for each subgroup. This process is repeated until the tree reaches the appropriate size, and outliers are detected using the finally configured tree. The Isolation Forest is a simple and computationally efficient, and that can quickly detect anomalies, even in large datasets. The proposed method can be applied to various types of data. An Isolation Tree can be applied to numerical and categorical data and to calculate outlier scores. The Isolation Tree calculates the outlier score for each data point, thereby measuring the degree of the outliers. However, Isolation Forests have certain drawbacks. The deeper the tree, the greater the likelihood of overfitting. The higher the density of the dataset, the lower the performance of the Isolation Tree. An Isolation Forest is primarily used to detect outliers and data showing unusual patterns. The performance of the outlier score calculation is used by the Isolation Forest to measure the degree of outliers because it can calculate the outlier score of each data point. This facilitates determining which data are most ideal in the dataset. Therefore, an Isolation Forest is used as a simple and efficient outlier detection algorithm that applies to various data types [44].

## 3. LSTM Autoencoder Based Anomaly Detection

### 3.1. Model Structure

As shown in Figure 4, PCA is a multivariate method based on quadratic statistical characteristics that convert a high-dimensional space into a low-dimensional space using coordinate transformation. PCA efficiently performs information compression and data correlation elimination [45]. Furthermore, reducing three-dimensional data to two dimensions simplifies the visualization of data and makes it easy to grasp the structure and pattern of data, because they can be visualized on two-dimensional planes. This helps in reducing data complexity and computational costs because it extracts the principal components that best preserve the characteristics of the data through dimension reduction. As the principal components are selected based on data variance, the reduction to two dimensions can represent the distribution of the original data. This makes it easier to understand trends and patterns in the data. After performing the process on both normal and abnormal data, autoencoder learning was performed using only the normal data.

The LSTM-AE was used to detect outliers in the time series data. The model compressed the input data into a low-dimensional latent space and reconstructed the original input data. In this case, the reconstruction loss between the reconstructed and the original data was calculated. The reconstruction loss is a value representing the difference between the input data and reconstructed data; the smaller the loss, the more accurate the reconstruction of the input data. These LSTM-AE models learn normal data, identify patterns in the corresponding data, and detect data with different patterns as abnormal values. Data with large reconstruction losses are classified as outliers because of the large differences between the original and reconstructed data. During model learning, the LSTM-AE was trained using only normal data, and the threshold for reconstruction loss was set. This threshold was set based on the distribution of reconstruction losses for normal data. The test data were evaluated using learned models and thresholds, and data with reconstruction losses greater than these thresholds were judged as outliers. The reconstruction loss of the LSTM-AE was used for outlier detection because the model was trained based on normal data. After the LSTM-AE learned the pattern of the normal data, the outliers had a different pattern from the corresponding pattern; thus, the reconstruction loss was large. Therefore, it is possible to detect outliers through the reconstruction loss, and the careful coordination of outlier detection is possible by adjusting the reconstruction loss threshold.

### 3.2. Anomaly Detector

The Isolation Forest and Autoencoder algorithms are used for outlier detection in unsupervised learning. These two algorithms focus on identifying and isolating outliers from the unlabeled data. The Isolation Forest uses data distribution to identify outliers. The algorithm assumes that outliers are rare and independent. The Isolation Forest operates by cutting the data, constructing a tree, and calculating an outlier score by measuring the length of the path in which each sample reaches the terminal node. This algorithm is faster than the autoencoder and has the advantage of being less affected by data dimensions. An autoencoder is a neural network-based algorithm that detects anomalies by minimizing reconstruction errors in the data. The autoencoder compresses the input data into a low-dimensional latent space and reconstructs the original input data. In this case, data with large reconstruction errors were considered as outliers. The autoencoder performed well in learning complex patterns of data and detecting outliers based on them and could be used more effectively to detect outliers by automatically adjusting the thresholds. LSTM-AE is a neural network-based algorithm suitable for time series or sequential data. LSTM is a type of RNN that can learn long-term dependencies [46].

The LSTM-AE uses LSTM cells to learn the complex patterns of the sequence data and because it can detect and remember long-term dependencies of sequential data, it is effective in detecting outliers in sequential data, such as time-series data or natural language processing. For outlier detection, the Isolation Forest and LSTM-AE use their respective methods to analyze the patterns and distributions of the data and identify outliers. Thus, they can be used to detect outliers and improve the data quality in unsupervised learning. LSTM-AE is a type of RNN that can learn long-term dependency, compress input data into a low- dimensional latent space, and then reconstruct the original input data. In this case, data with large reconstruction errors were considered as outliers. LSTM can detect and remember the long-term dependencies of sequential data. Therefore, the LSTM-AE is effective in detecting outliers in sequential data, such as time-series data or natural language processing [47].

The anomaly detector is judged based on the threshold set by the normal data of LSTM-AE, and if the calculated threshold of each data point is higher than the threshold of the normal data, the data point is judged as an anomaly.

### 3.3. Loss Function

The proposed LSTM-AE model detects outliers that are outside the normal range. As we performed unsupervised training, we used only normal vibration data to train the autoencoder, which requires that the training data consist of normal data that does not contain outliers. When dimensionality reduction is performed, the reconstruction error is calculated, and in this study, we used the mean squared error (MSE) loss function [46,48]. MSE is defined as the average of the squared differences between the actual and predicted values. A smaller MSE value indicates a higher prediction accuracy of the model. The MSE used for reconstruction error is defined as follows.
(1)$$\frac{1}{n}\sum_{i=1}^{n}{\left(X_{i}-{\hat{X}}_{i}\right)}^{2}$$

The optimal threshold value is the point where the difference between precision and recall is minimized. There are several loss functions as evaluation indicators, and MSE squares the loss between the actual value and the predicted value and imposes a panel to enable the model to make accurate predictions. The receiver operating char curve (ROC) is used to evaluate the accuracy of outlier detection, visualize the performance of the binary classification models, and for outlier detection. The ROC curve has a True Positive rate (=sensitivity) on the y-axis and a False Positive rate (=1-specificity) on the *x*-axis. TPrate and FPrate are defined as follows [49].
(2)$$TPrate=\frac{TP}{AP}\;\mathrm{and}\;FPrate=\frac{FP}{AN}$$

The ROC curve represents the performance of the classification model at all thresholds and is a graph showing the change in the sensitivity and specificity of outlier detection when the threshold of reconstruction error changes. The closer the area under the ROC curve (AUC) value is to 1, the better the performance of the model. The minimum value of AUC is 0.5, which means that the model cannot classify classes. In this study, the AUC was calculated using an ROC curve to evaluate the outlier detection performance of the LSTM-AE model, and the accuracy of the model was confirmed [50].

## 4. Experiment and Results

### 4.1. Experiment Environments

The development environment of this study is as follows. OS: Mac OS v13.3.1; Python: Python v3.10.12; CPU: AMD EPYC 7B12; Memory: 13,294,252 kB.

Data were collected from a wind farm in Northern Sweden. The wind turbines were of the same model and included an integrated state-monitoring system that transmitted data to a state-monitoring database, from which the vibration data used in this study were accessed. Each wind turbine had a three-speed gearbox, including two sequential planetary gear stages and a helical gear stage. Each gearbox had four accelerators near the different gear stages. All measurement data corresponded to the axial direction of the accelerometer mounted on the output shaft bearing housing of each turbine. The sampling rate was 12.8 kHz, and the length of each signal segment was 1.28 s (16,384 samples). Signal segments were approximately 12 h apart for 46 consecutive months over the past 10 years [51].

Figure 5 shows the components of each stage, including the support bearing are shown. In this paper, data from one wind turbine with two bearing failures were used as fault data. The illustration highlights the position of the defective bearing.

### 4.2. Parameter Setting

The number of units in the LSTM of the encoder portion was set to 515 and 256. Increasing the number of units in the LSTM model allowed it to learn more complex patterns. More units could be used to process more complex time-series data by increasing the capacity of the model. LSTM can learn long-term dependencies, and this model is used to perform reconstruction work on time-series data. Increasing the complexity of the time-series patterns learned makes the model more effective in identifying patterns in normal data and detecting outliers. The learning rate was set as 0.0001. The learning rate determines the rate at which the model updates its weights. Therefore, selecting an appropriate learning rate allows the model to converge quickly to the optimal solution, and determine the optimal loss value. The learning rate determines the weight update size. A learning rate that is too small can slow the learning speed, whereas a learning rate that is too large can make the learning unstable. Therefore, selecting an appropriate learning rate is important. The Adam optimizer combines the ideas of the momentum and RMSProp algorithms to automatically adjust the learning rate used for the weight updates. The Adam optimizer estimates the mean and variance of the gradient using the moving average and performs weight updates accordingly. Thus, the Adam optimizer performs well with various types of data and models. When the optimizer was set to Adam in the compilation step, the weight of the model was updated using the Adam optimizer generated above. In addition, when set to loss, the MSE was used as the loss function. The MSE is a value averaged by squaring the difference between the actual and predicted values and is used to measure the reconstruction loss of the model. The goal is to restore the input data as effectively as possible by minimizing the reconstruction loss [52].

### 4.3. Comparison of Normal and Abnormal Data

Figure 6 and Figure 7 represent the waveforms of the normal and abnormal data. Considering the raw signal alone, it was difficult to discern the differences between the two datasets. Thus, the waveform was analyzed by decomposing the signal using various frequency bands through the WPT.

Therefore, by examining the WPT results shown in Figure 8 and Figure 9, the waveform in each frequency band and the difference in the frequency band between normal and abnormal data could be identified. Because the characteristics of the waveforms were different for each frequency band, abnormal data could be identified, or the cause could be inferred by analyzing the characteristics.

In WPT the sub-bands were generated for each level, and the decomposition at each level was performed using low- and high-pass filters. Each frequency band is represented by a sub-band and the waveform can be examined by selecting a sub-band of a specific frequency band from the given data. In this study, we analyzed the waveforms of the normal and abnormal data by selecting the eighth bandwidth that could be selected by analyzing the given data. This implies that the information related to the characteristics of the corresponding frequency bands must be obtained. When the waveforms of normal and abnormal data were analyzed at the eighth bandwidth, the characteristics and differences in the corresponding frequency bands were confirmed. This facilitated comprehending the difference in frequency bands between normal and abnormal data and discovering unusual patterns or abnormalities in the abnormal data.

Figure 8 Bandwidth of the eighth WPT at normal frequency. Figure 9 Bandwidth of the eighth WPT with abnormal data. In this experiment, 16 waveforms of normal and abnormal data were visualized, which enabled us to visually identify the differences and the singularities between the two datasets. Figure 8 and Figure 9 show the normal and abnormal waveforms of the eighth bandwidth, respectively, that represented the characteristics of the data among the visualized waveforms.

When PCA was performed using the waveform data of the selected bandwidth, the distribution of the data and major fluctuation factors were confirmed. PCA helps highlight key patterns and differences in the data. Therefore, by comparing the PCA results of the normal and abnormal data, the difference and singularity between the two datasets were confirmed. The PCA results for the normal data showed the main patterns and characteristics of the waveform data for the corresponding bandwidth. This represented the consistent behavior of normal data that fits the bandwidth. However, the PCA results of abnormal data represented an outlier or a unique pattern. This indicated unexpected behavior or unusual characteristics in the waveform data of the corresponding bandwidth. Accordingly, data with a singularity were identified by comparing the PCA results of normal and abnormal data. Subsequently, abnormal values or operations occurring in the waveform data of the corresponding bandwidth may be detected. Because PCA is a powerful tool for performing dimensionality reduction and characteristic extraction of data, it facilitated the identification of singularities in data related to bandwidth.

Figure 10 is normal data after HPF and Figure 11 is abnormal data HPF. The HPF enhances the high-frequency signals of normal and abnormal data, removes the low-frequency components from the frequency domain making the components of the signal clearer, and emphasizes the high-frequency components to achieve clearer results. HPF is mainly used in signal or image processing and is useful only when observing components above a specific frequency. HPF can be applied to extract and analyze the desired frequency band. This helps in identifying important frequency components of the data. In addition, removing low-frequency components from the image and highlighting high-frequency components allows for better observation of the details, boundaries, and noise in the image. HPF performs filtering on a given signal or image in the frequency domain; therefore, any component below that frequency is removed and any component above that frequency remains highlighted. This allows the analysis of the frequency characteristics of the data or the extraction of information contained in a specific frequency band.

Figure 12 shows a visualization of the bandwidth of normal vibration eighth. Figure 13 shows a visualization of the bandwidth of abnormal vibration eighth. A three-dimensional space was established for the data visualization of a specific bandwidth obtained by performing WPT. First, the variables used for visualization were selected from the dataset to be analyzed. The selected variable corresponds to the axis of the three-dimensional space. The range of each axis was set according to the selected variables. This was determined by considering the minimum and maximum values of the selected variables. Three-dimensional space was used to visualize data based on selected variables. Each data instance was mapped to a specific coordinate in space according to the selected variables. Through this, the distribution and pattern of the data were understood intuitively and analyzed. Figure 12 and Figure 13 are the visualization data mapped onto 3D data with the time and amplitude of each data point.

Table 1 shows the variance of the data when the dimensions are reduced by the number of corresponding main components. Normal data accounted for 98.9% of the total distribution of data, and abnormal data accounted for 98.5% of the total distribution of data. This ratio aided in selecting the main ingredient. If the principal components were selected in the order in which the Explained Variation Ratio of the principal components was large, the upper principal components that explain most of the variance in the data could be selected. This allowed us to preserve the important features of the data even if the dimensions were reduced. Checking the Explained Variation Ratio from the principal component analysis results helped us evaluate how well we described the variance of the dataset and determined the appropriate level of dimensionality reduction.

Figure 14 and Figure 15 are visualization graphs that confirm the distribution of data by performing PCA. This process facilitated a better understanding of the trends and patterns in data, and it could be observed that normal and abnormal data have different characteristics and patterns.

By representing data in low dimensions using PCA, the distribution and clustering of normal and abnormal data were visually understood. Normal data formed denser clusters, while abnormal data appeared as distinct clusters or isolated data points. The main component obtained from the PCA was the axis that best described the variability of the original data. Therefore, the important data characteristics could be identified through the main components, and the differences between normal and abnormal data could be visualized and confirmed. Based on the value of the main component or projection value of the data, abnormal data were interpreted as deviating from a general pattern.

Figure 16 and Figure 17 are the visualization results of dimension reduction with the main components obtained through the Explained Variation Ratio. Notably, the number of vibration components decreased because dimensional reduction was performed. Compared with Figure 10 and Figure 11, it can be seen that the characteristics of the data are alive.

Figure 18 and Figure 19 show the results of detecting outliers and applying the Isolation Forest using the preprocessed data in unsupervised learning. Although there was a difference in the three-dimensional distributions of Figure 18 and Figure 19 with abnormal data, an accurate classification was not made in terms of the number.

The Isolation Forest works by cutting off the data. The data were divided using a randomly selected segmentation plane, which was repeated to measure the path length through which each data point reached the terminal node of the tree. Outliers tended to have short path lengths, and the outlier score was calculated based on the path length. Because outliers have paths that differ from the general data, they were relatively more likely to have shorter paths. Therefore, even if an Isolation Forest was used in a dataset composed only of normal data, data with a shorter path than the main component were classified as abnormal values. Isolation Forest, which detected anomalies based on the length of the data path, classified anomalies even in datasets composed of only normal vibration data; therefore, it could be used in combination with other anomaly detection algorithms or other techniques, or additional preprocessing or feature extraction methods. In addition, the performance of the algorithm could be evaluated and improved by checking the evaluation index of the classification results.

### 4.4. Results

We conducted outlier detection using an LSTM-AE to utilize more sophisticated artificial intelligence models. Figure 3 shows the LSTM was compressed with a model in which only normal data were learned and then reconstructed into the form of the original data. When compressed, the data characteristics were extracted, and the loss rate was the reconstruction error rate. An outlier was detected by the difference between the input and output data, and if the reconstruction error rate exceeded the threshold calculated by the model, it was considered an outlier. There were two layers of LSTM in the encoder and two layers in the decoder.

Considering visualization, Figure 20 shows that data processing using the method proposed in this paper makes good use of the characteristics of the most original sound. It was performed in three order: the method of implementing only PCA in the original data, the method of implementing PCA after performing HPF, and the method of implementing PCA through WPT and HPF.

Detailed descriptions of each figure are as follows: (a) Only by PCA on the original raw data, (b) PCA-only data into the LSTM-AE in the original raw data, (c) HPF on the original data and then PCA, (d) HPF on the original data and then PCA performed data into the LSTM-AE, (e) data performed by performing WPT on the source data, HPF at a specific bandwidth, and then PCA, (f) data regenerated by performing WPT on the original data, performing HPF at a specific bandwidth, and then putting the PCA data into the LSTM-AE.

Table 2 shows the model loss based on MSE for each epoch from a minimum of 30 to a maximum of 150 epochs. It can be seen that the WPT-HPF-PCA method, which is the preprocessing method proposed in this study, has the lowest loss.

Figure 21 shows the results of the experiments using models of the same structure. Notably, the more preprocessed the data were, the closer the AUC was to 1.

Detailed descriptions of each figure are as follows: (a) Only by PCA on the original raw data, (b) PCA-only data into the LSTM-AE in the original raw data, and (c) HPF on the original data and then PCA.

The dataset preprocessing method proposed in this paper showed substantially better results than the raw data. Abnormal detection was performed using raw data. It can be seen that an accuracy of 94.10% was obtained from Table 3, and a higher accuracy of 97.44% was achieved when the pretreatment method proposed in this study was applied. Based on these results, we observed the effect of maximizing the difference between normal and abnormal data in the process of extracting and strengthening data characteristics using WPT and HPF. Through WPT, the data was decomposed into various frequency bands so that the characteristics of the data could be better understood. Afterward, HPF was applied to maximize the difference while strengthening the characteristics of the data. Finally, through principal component analysis, computing could be efficiently performed, and outliers could be identified and classified. Therefore, the dataset preprocessing method proposed in this study shows a higher performance than with raw data, which could substantially improve the accuracy of anomaly detection by extracting and strengthening the characteristics of the data.

In Figure 22, (a) is a visualization of the distribution of the raw data based on the thresholds set by the model trained on the raw data; (b) is a visualization of the distribution of the data that performed WPT-PCA based on the thresholds set by the model trained on the data that performed WPT-PCA; (c) is a visualization of the distribution of data that performed WPT-HPF-PCA based on the thresholds set by the model trained on the data that performed WPT-HPF-PCA.

The thresholds shown in Figure 22 have different values. This is because the model calculated the optimal thresholds with data processed by different preprocessing methods; (a) shows that there is a relatively large mix of normal and abnormal data within the thresholds, but (c) shows that the abnormal data within the thresholds has decreased significantly compared to (a); (c) shows that the WPT-HPF-PCA performed as the final preprocessing in this study resulted in almost no abnormal data entering the thresholds. This visualization of the data distribution indicates that the model has a good understanding of the reconstruction loss rate threshold for normal data.

### 4.5. Discussion

The outlier detection performed after the proposed preprocessing method outperformed the outlier detection performed with conventional data. Preprocessing techniques that combine WPT, HPF, and PCA are useful in a variety of data analysis and signal processing applications. Each technique contributes to extracting important characteristics from the data, reducing noise, and reducing dimensionality to facilitate the interpretation of the data. WPT can analyze signals in both the time and frequency domains, allowing important characteristics to be extracted from even complex nonlinear signals. This allows it to better capture information occurring in different frequency bands of the signal. HPF allows you to remove low-frequency noise. This helps to emphasize the high-frequency components of the data, the parts of the data that have the most variation. As mentioned above, both WPT and HPF contribute to reducing noise, and HPF, in particular, can improve the quality of your data by effectively removing low-frequency noise. PCA reduces the dimensionality of the data, preserving key characteristics that are less sensitive to noise. In the process, unnecessary information is removed, improving the signal-to-noise ratio of the data. By reducing the dimensionality of the data with PCA, the amount of computation required for analysis or modeling is reduced. This can save time and resources, especially when dealing with large datasets. Combining these three techniques highlights important characteristics in the data, reduces noise, and reduces dimensionality. As a result, the structure of the data becomes clearer and it becomes easier to identify important patterns or relationships. LSTM, the model used for outlier detection, is very effective at learning long-term dependencies in time series data. By capturing important temporal patterns and dynamic changes in preprocessed data, they can learn temporal characteristics that are essential for outlier detection. The autoencoder (AE) have the ability to efficiently compress (encoder) and reconstruct (decoder) input data. By combining with LSTMs, AEs can learn complex patterns in time series data and reconstruct the structure of normal data based on them. Outliers are characterized by large differences between the normal data structure and the reconstructed data, and this difference can be used to detect outliers. The use of LSTM-AE models for outlier detection provides a powerful way to understand complex data structures, including temporal patterns, based on high-quality preprocessed data, and to effectively identify data points that fall outside the normal range.

Previous studies have required human intervention or had high computational complexity, making them difficult to apply to real-world scenarios. However, the LSTM-AE model used in this study presents a lightweight algorithm that can detect anomalies in real-time and respond immediately. This is a great advantage, especially for preventing unexpected failures in critical infrastructures such as wind turbines. Many anomaly detection studies are based on supervised learning, which encounters the need for sufficient labeled data. In contrast, the proposed method adopts an unsupervised learning approach which can be applied to unlabeled or expensive-to-label data. This is particularly beneficial in real-world industrial settings. In unsupervised learning, it is difficult to evaluate the performance of a model or generalize to other datasets because no labels exist for the data. This limitation is expected to be a useful strategy if labeling can turn unsupervised data into supervised learning. Self-supervised learning using labeled datasets using a classification method with automatic labeling is expected to produce a variety of models. The proposed method can also perform successful outlier detection in noisy and complex vibration data from industrial environments. Unsupervised learning has the advantage of being able to discover the inherent structure of a dataset, but it has the disadvantage of being generally inconsistent and difficult to interpret. In addition, fault classification needs to be performed using datasets containing different types of fault data in order to classify and predict fault types. To demonstrate the superiority of this research, more model comparisons are needed, and we plan to combine them with XAI to obtain reliable results. Also, since this research was conducted on a limited dataset, we plan to demonstrate its validity on various datasets and real-world industrial environments. Continued technical and research efforts to address these issues are expected to open up new possibilities and contribute to the broader field of artificial intelligence.

## 5. Conclusions

In our proposal, we explored the preprocessing of normal data sets using the WPT-HPF-PCA method and the detection of outliers using the LSTM-AE model. The key idea was to transform the data using WPT and HPF and perform dimensionality reduction using PCA to extract the characteristics of the data. In this study, we first decomposed the data using WPT and applied HPF to extract the signals in the desired frequency range. We then performed PCA to reduce the dimensionality of the data and trained the LSTM-AE model accordingly. The LSTM-AE compressed the input data into a low-dimensional latent space to restore the original data. The trained model was used to calculate the reconstruction loss for normal data, which was used to find the threshold to detect outliers. In the experiments, we trained and evaluated the model using different normal datasets with sequential step preprocessing, and evaluated the performance of the model by comparing the reconstruction loss with the actual outliers. The experimental results showed that the proposed method performed well in detecting outliers in the test data using the threshold obtained by training on normal data only, and achieved good results compared to other outlier detection preprocessing algorithms.

## Figures and Tables

**Figure 1 sensors-24-02833-f001:**
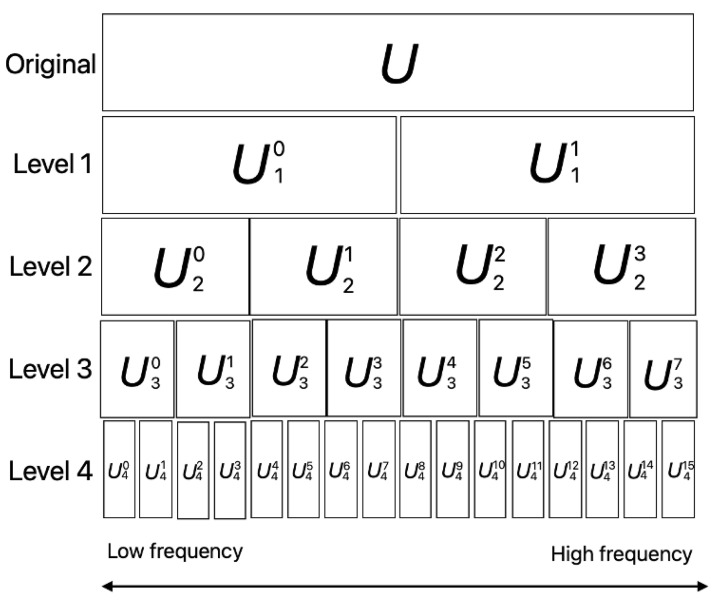
Wavelet Packet Transform.

**Figure 2 sensors-24-02833-f002:**
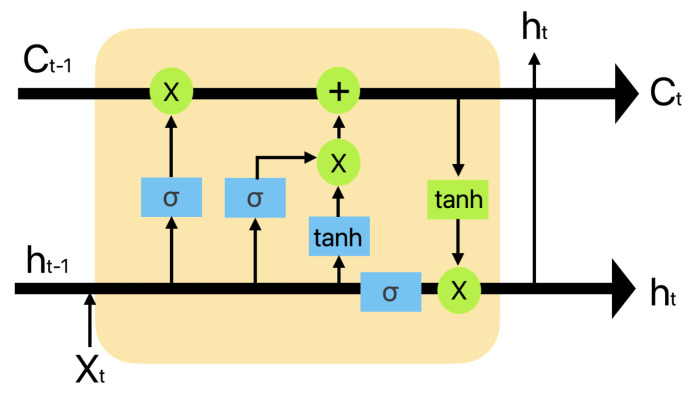
LSTM cell.

**Figure 3 sensors-24-02833-f003:**
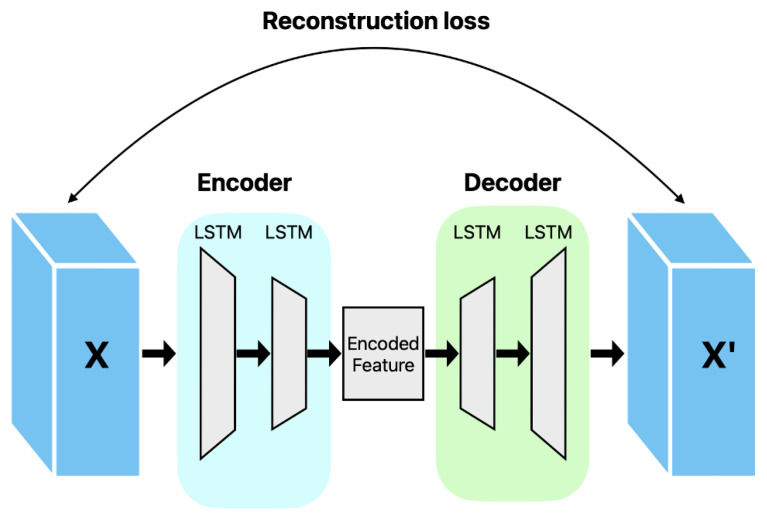
LSTM Autoencoder.

**Figure 4 sensors-24-02833-f004:**
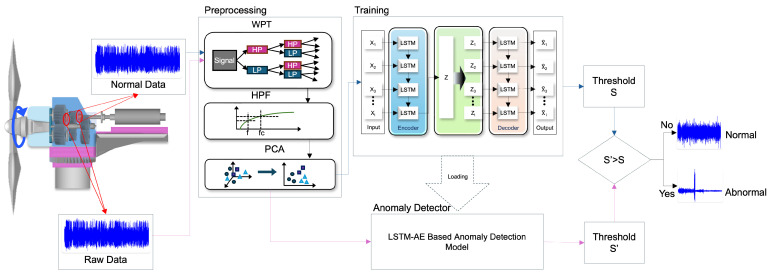
The Proposed Model Framework.

**Figure 5 sensors-24-02833-f005:**
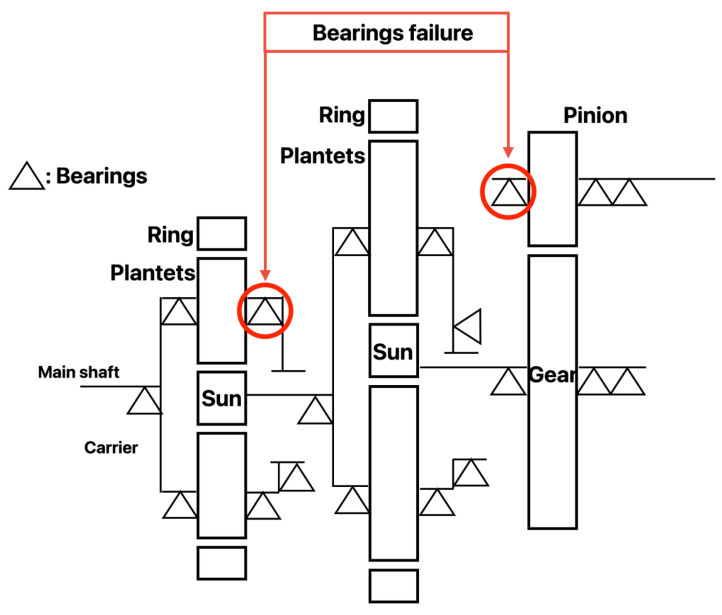
Gearbox Schematic for Wind Turbines.

**Figure 6 sensors-24-02833-f006:**
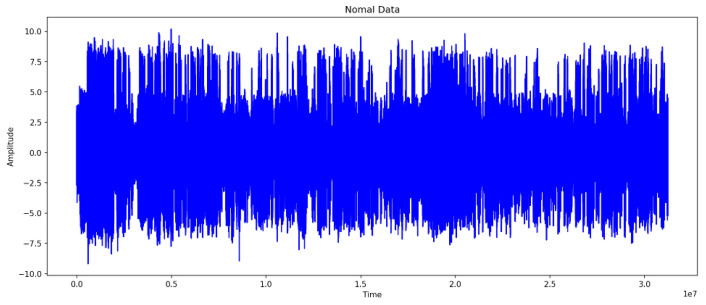
Wave of Normal Data.

**Figure 7 sensors-24-02833-f007:**
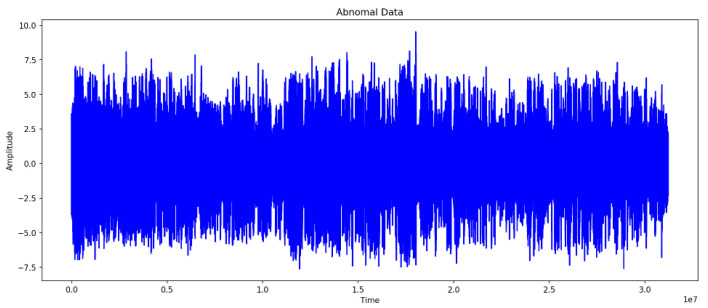
Wave of Abnormal Data.

**Figure 8 sensors-24-02833-f008:**
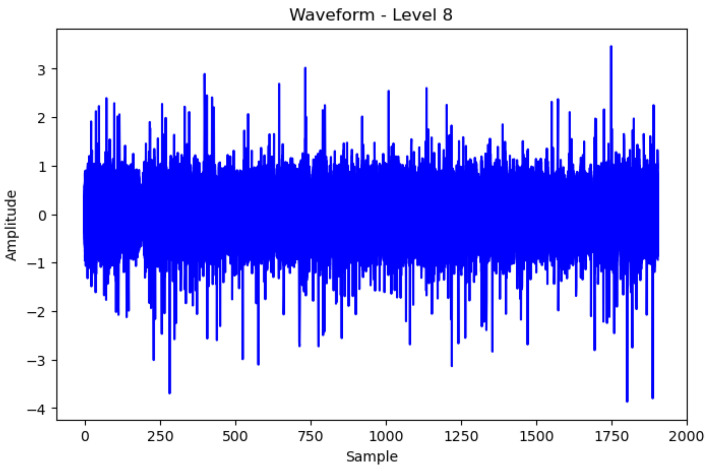
Wave of Normal WPT eighth Data.

**Figure 9 sensors-24-02833-f009:**
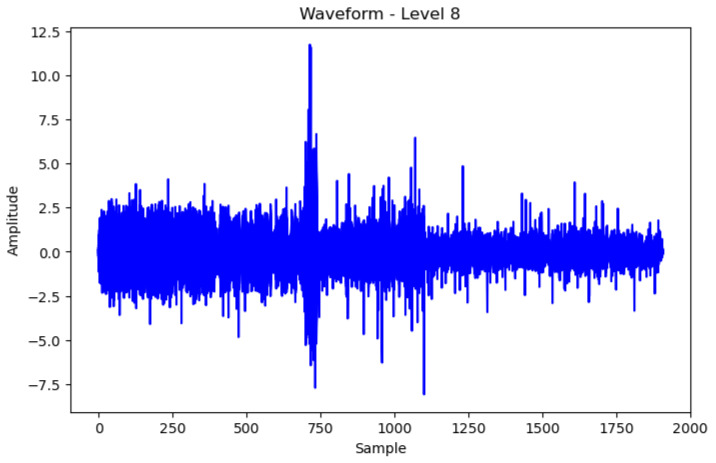
Wave of Abnormal WPT eighth Data.

**Figure 10 sensors-24-02833-f010:**
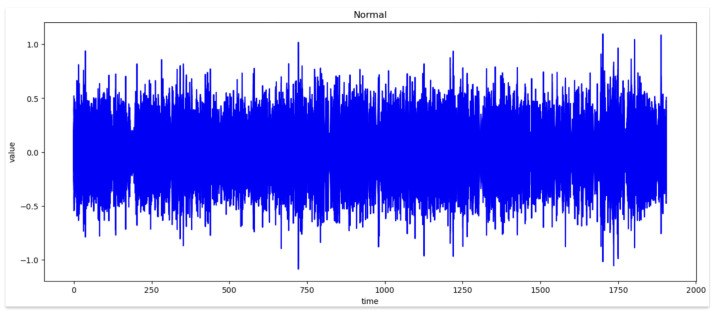
HPF of Normal Data.

**Figure 11 sensors-24-02833-f011:**
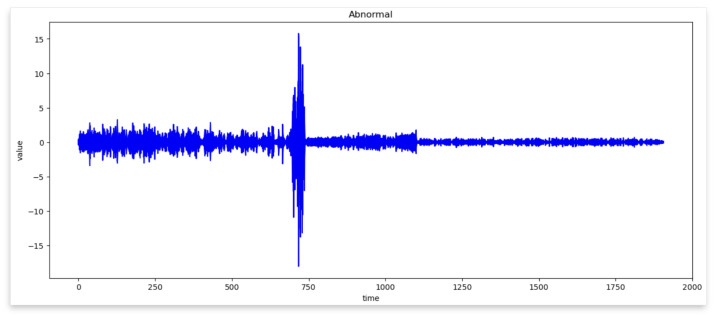
HPF of Abnormal Data.

**Figure 12 sensors-24-02833-f012:**
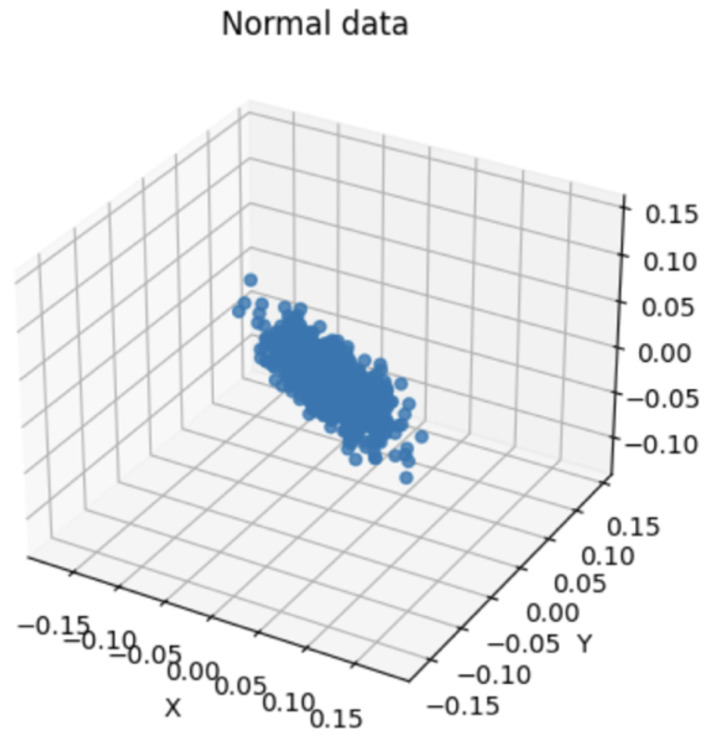
Bandwidth of Normal Vibration eighth.

**Figure 13 sensors-24-02833-f013:**
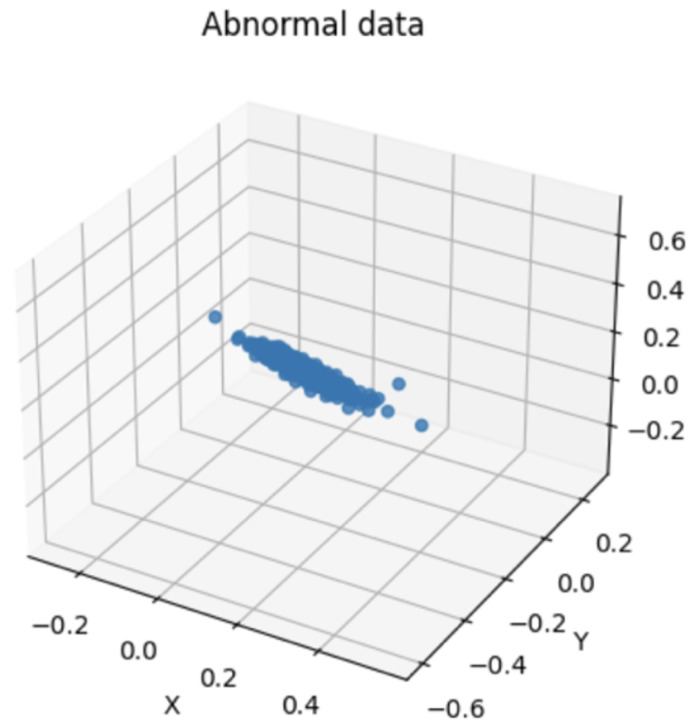
Bandwidth of Abnormal Vibration eighth.

**Figure 14 sensors-24-02833-f014:**
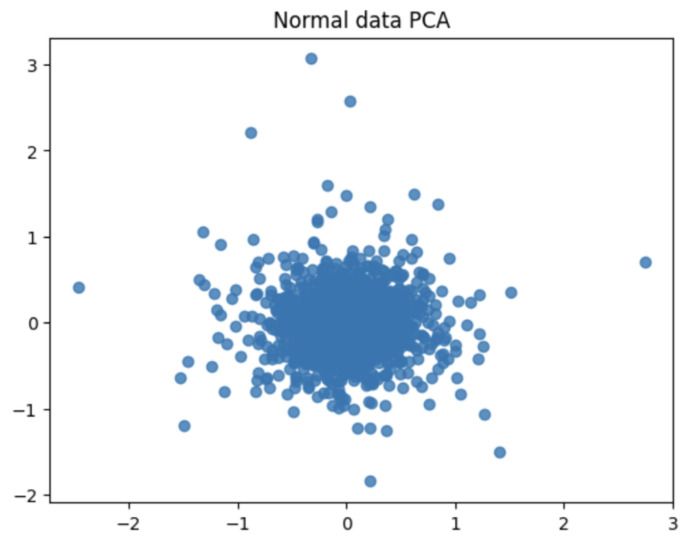
PCA of normal data.

**Figure 15 sensors-24-02833-f015:**
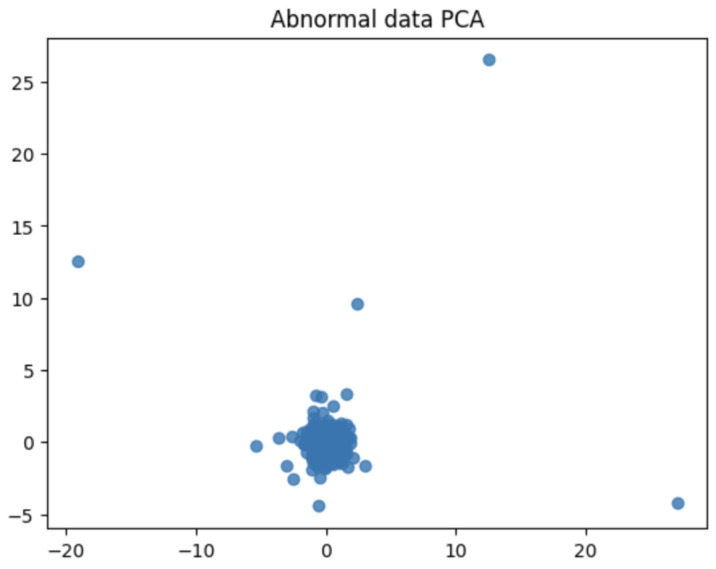
PCA of abnormal data.

**Figure 16 sensors-24-02833-f016:**
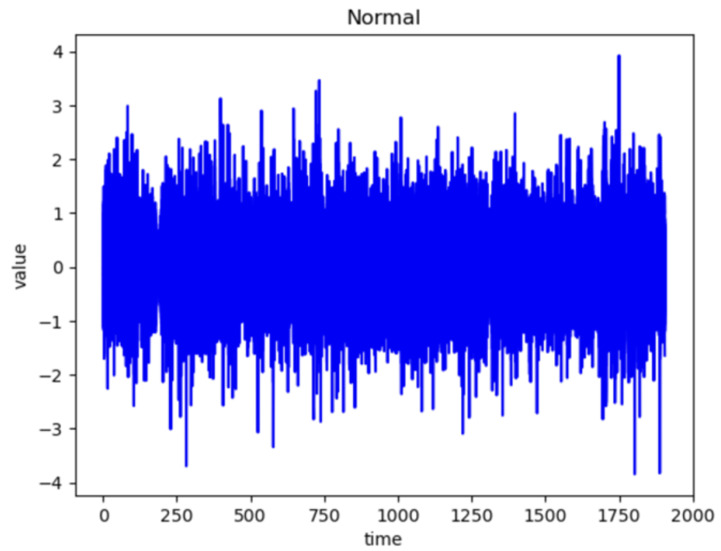
Dimension reduction for PCA of normal data.

**Figure 17 sensors-24-02833-f017:**
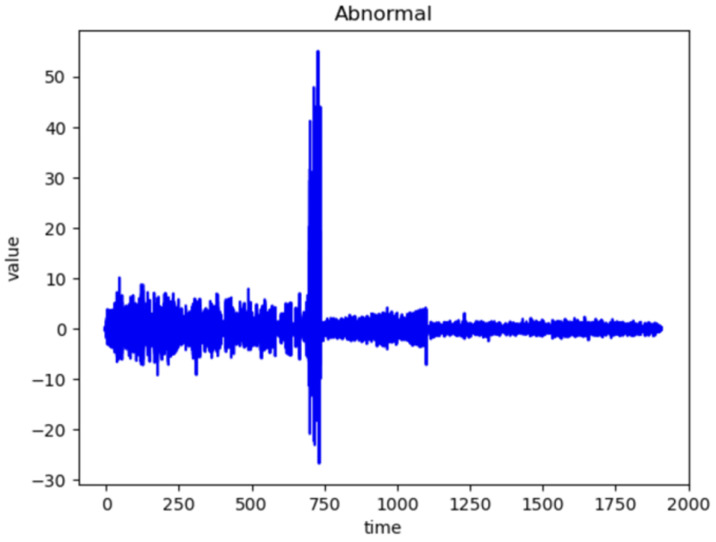
Dimension reduction for PCA of abnormal data.

**Figure 18 sensors-24-02833-f018:**
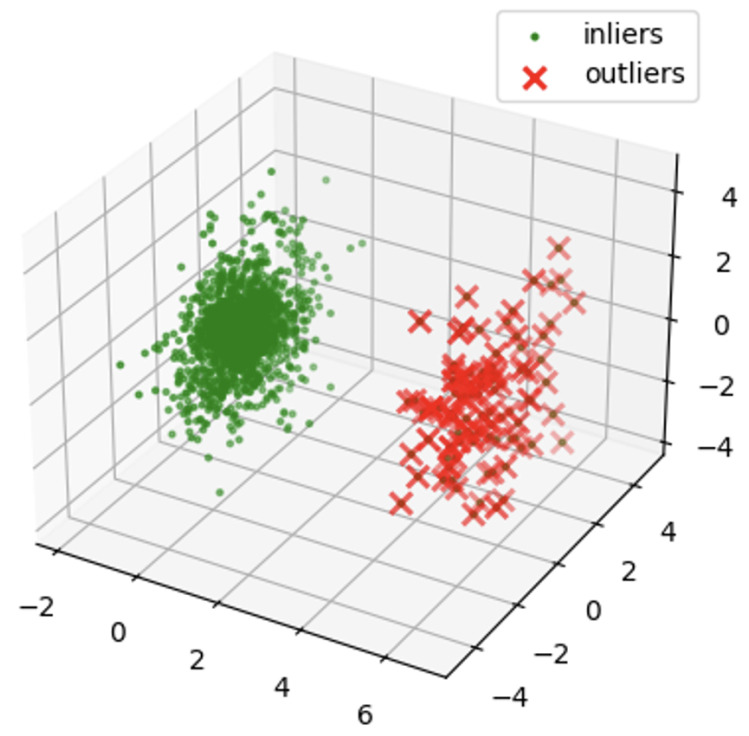
Isolation Forest of normal data.

**Figure 19 sensors-24-02833-f019:**
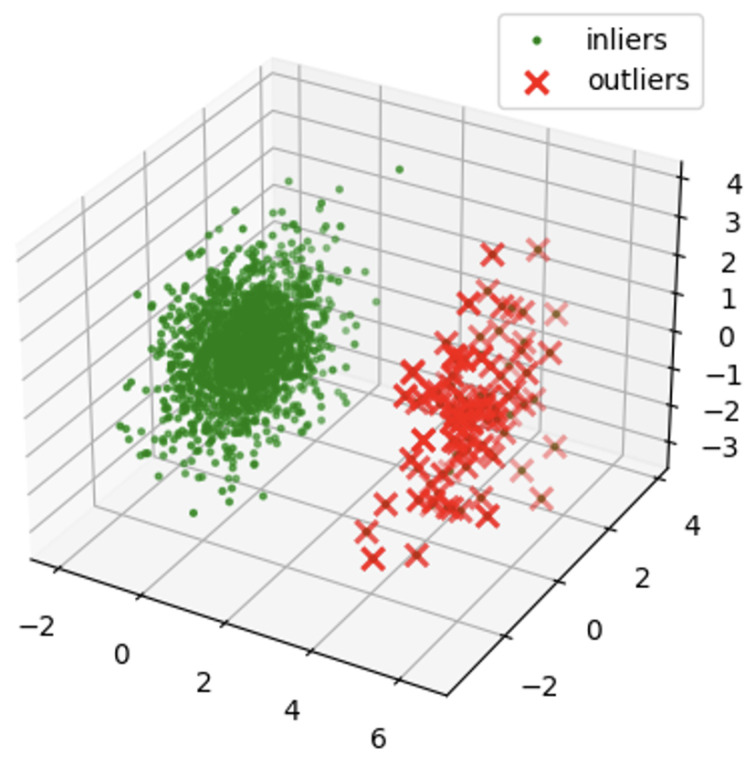
Isolation Forest of abnormal data.

**Figure 20 sensors-24-02833-f020:**
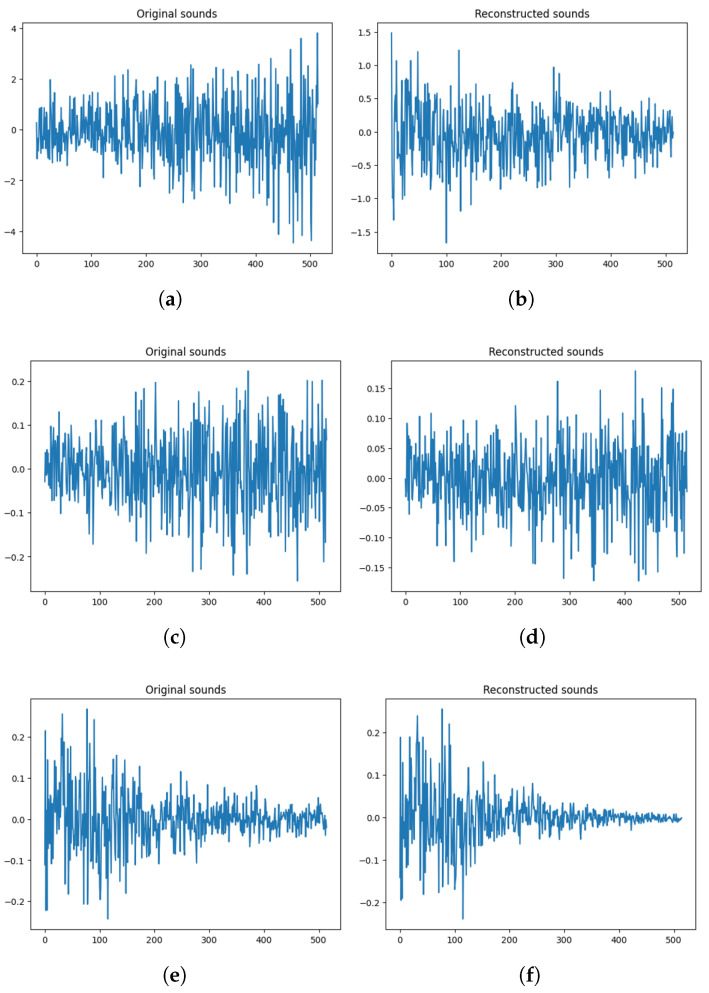
Visualizations of each preprocessing. (**a**) PCA of raw original data; (**b**) PCA of raw reconstructed data; (**c**) HPF-PCA of original data; (**d**) HPF-PCA of reconstructed data; (**e**) WPT-HPF-PCA of original data; (**f**) WPT-HPF-PCA of reconstructed data.

**Figure 21 sensors-24-02833-f021:**
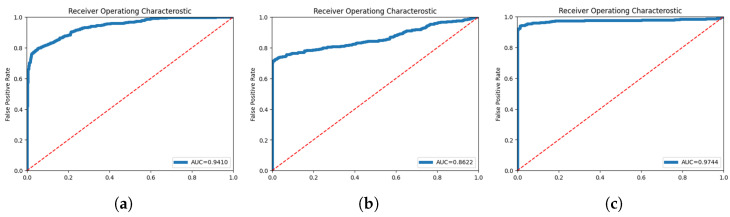
AUC of each data. (**a**) PCA data; (**b**) WPT-PCA data; (**c**) WPT-HPT-PCA data.

**Figure 22 sensors-24-02833-f022:**
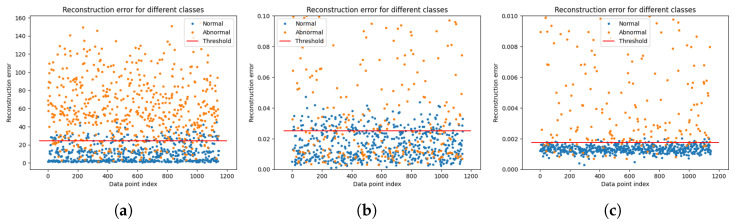
Distribution and threshold of each data. (**a**) PCA data; (**b**) WPT-PCA data; (**c**) WPT-PCA-HPF data.

**Table 1 sensors-24-02833-t001:** Data Explained Variance Ratio.

	Normal Data	Abnormal Data
Components = 515	0.9890	0.9856

**Table 2 sensors-24-02833-t002:** Model loss.

	PCA Data	HPF-PCA Data	WPT-HPF-PCA Data
Epoch 30	35.0556	0.1078	0.0065
Epoch 60	24.8907	0.0604	0.0029
Epoch 90	18.6556	0.0333	0.0021
Epoch 120	14.2359	0.0220	0.0017
Epoch 150	11.7810	0.0181	0.0015

**Table 3 sensors-24-02833-t003:** AUC Value.

	PCA	HPF-PCA	WPT-HPF-PCA
AUC	0.9410	0.8622	0.9744

## Data Availability

S. Martin del Campo Barraza, F. Sandin, and D. Strömbergsson, ‘Dataset concerning the vibration signals from wind turbines in northern Sweden’. 2018. (https://ltu.diva-portal.org/smash/record.jsf?pid=diva2%3A1244889&dswid=2589 accessed on 3 September 2018).

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
