# Peer review of "LSTM-Autoencoder Based Anomaly Detection Using Vibration Data of Wind Turbines"

_sensors, 2024, doi:10.3390/s24092833_

Round 1
Reviewer 1 Report
Comments and Suggestions for Authors
1.According to the references, the novel contribution of the paper is not obvious enough. The authors should precisely illuminate the differences between their work and the prior studies.
2.Please elaborate on the definition in Section 1.2.
3.In Section 4, it is suggested to add comparative experiments to highlight the superiority of the proposed method.
4.Only a numerical examination has been provided, it is recommended to validate the method with a real case.
5.In Section 5, discussions are weak.
6.The discussion and conclusions can be divided into two sections.
7.Please enrich the management insights of this paper.
8.Please unify the format of the manuscript.
Comments on the Quality of English Language1.Please recheck the paper writing, and correct the typo and grammatical errors in the paper.
Author Response
Please see the attached for your review and comments.

Reviewer 2 Report
Comments and Suggestions for Authors
(1) The introduction section should incorporate the latest research findings on abnormal detection of Wind Turbines, along with outlining the gaps and advantages of the current study.
(2) The related work section and subsequent analysis should avoid redundant explanations of basic and common scientific analysis methods. Consider revising and enhancing the entire article by removing unnecessary introductions.
(3) The paper primarily employs unsupervised learning methods to predict anomalous data in wind turbines, such as the Isolation Forest and Autoencoder algorithms. However, it provides normal data within the paper; how are these normal data determined? At the same time, the threshold s' is established using the normal data, which essentially constitutes supervised learning.
(4) Verify the correct threshold calculation formula used in Figure 4, as it appears inconsistent with Formula 1. Additionally, ensure that the interpretation of Figure 4 is accurate, stating that data is normal when s > s' and abnormal when s < s'.
(5) Provide an explanation for the significant variance in vertical coordinate reconstruction error depicted in Figure 22.
(6) Revise the conclusions to include discussions on future research directions and limitations of the study. It is advisable for the authors to present arguments related to these aspects.
Comments on the Quality of English Language(1) Please meticulously review the paper for errors and redundancies, especially in abstract.
Author Response

(The authors gave the same response as above.)

Reviewer 3 Report
Comments and Suggestions for Authors
In this manuscript, the authors develop a methodology for identifying anomalies in the patterns of wind turbines vibration in order to pro-actively detect malfunction circumstances that may ultimately lead to breakdown. In specific, the proposed method includes the decomposition of the time-series signal using wavelets and then specify the frequency band (also using high pass filters) that exhibits large difference amongst the normal and abnormal data. Furthermore, the authors utilize the principal component analysis to identify the signal components that play a significant role, and they use the pre-processed normal data to train a long short-term memory (LSTM) Autoencoder for purposes of outlier detection. Finally, the proposed method is applied to data collected from a wind farm, demonstrating the performance of the developed algorithm. The paper is in general interesting and has technical value. Some minor comments:
Please improve the quality of Figure 10.
Figures 12-15, 18, 19 – please explain better what the axes of these figures are and how the units of the axes are calculated.
Finally, I could not find anywhere the temporal dimension related to the LSTM model, e.g., what is the time window that you use for predicting the anomaly? It would be beneficial to briefly elaborate on this issue, also considering time-series prediction methods applied in scientific diverse domains, e.g.:
1. "A supervised deep learning framework for proactive anomaly detection in cloud workloads." 2017 14th IEEE India Council International Conference (INDICON). IEEE, 2017.
2. ReRe: A lightweight real-time ready-to-go anomaly detection approach for time series." 2020 IEEE 44th Annual Computers, Software, and Applications Conference (COMPSAC). IEEE, 2020.
3. "Proactive anomaly detection for robot navigation with multi-sensor fusion." IEEE Robotics and Automation Letters 7.2 (2022): 4975-4982.
4. "Intelligent Mission Critical Services over Beyond 5G Networks: Control Loop and Proactive Overload Detection." 2023 International Conference on Smart Applications, Communications and Networking (SmartNets). IEEE, 2023.
5. "Achieving predictive and proactive maintenance for high-speed railway power equipment with LSTM-RNN." IEEE Transactions on Industrial Informatics 16.10 (2020): 6509-6517.
Comments on the Quality of English LanguageMinor typos and/or syntax errors are present in the manuscript, e.g., abstract line 6, line 16. Please carefully proofread your manuscript.
Author Response

(The authors gave the same response as above.)

Round 2
Reviewer 1 Report
Comments and Suggestions for Authors
1.The discussions in the Experiment and Results section are still weak.
2.The pictures in the paper are too simple, and it is suggested to add more descriptive information to the chart, such as the obvious difference between the pictures.
3.It would make more sense to have a consistent name format for Figures 12 and 13.
Comments on the Quality of English LanguageFine
Reviewer 2 Report
Comments and Suggestions for Authors
(1)In Figure 4, it is recommended to use arrows of different colors to represent the sequence of processing for normal data and raw data, thereby facilitating a more comprehensive understanding of the data processing flow and methodology.
(2)In figure 22(a), the vertical axis shows a reconstruction error value of over 100, while in figures (b) and (c) the values are in the range of 10^-2 and 10^-3. Why is there such a significant difference? Was the analysis conducted using different original data or the same original data?
Comments on the Quality of English Language
none
